# Facile and Selective Determination of Total Phthalic Acid Esters Level in Soft Drinks by Molecular Fluorescence Based on Petroleum Ether Microextraction and Selective Derivation by H_2_SO_4_

**DOI:** 10.3390/molecules27134157

**Published:** 2022-06-28

**Authors:** Enqin Xia, Ting Yang, Xuan Zhu, Qing Jia, Jun Liu, Wenlong Huang, Jindong Ni, Huanwen Tang

**Affiliations:** Dongguan Key Laboratory of Environmental Medicine, School of Public Health, Guangdong Medical University, Dongguan 523808, China; xiaenqin@gdmu.edu.cn (E.X.); yangting@gdmu.edu.cn (T.Y.); zhuxuan@gdmu.edu.cn (X.Z.); qingjia@gdmu.edu.cn (Q.J.); liujun@gdmu.edu.cn (J.L.); huangwenlong@gdmu.edu.cn (W.H.)

**Keywords:** phthalic acid esters, determination, petroleum ether microextraction, molecular fluorescence, soft drinks

## Abstract

Determining the level of phthalic acid esters (PAEs) in packaged carbonated beverages is a current need to ensure food safety. High-selectivity and -accuracy identification of individual PAEs can be achieved by chromatographic and mass spectrometric (MS) techniques. However, these methods are slow; involve complicated, expensive instruments in professional laboratories; and consume a large amount of organic solvents. As such, a food analysis method is needed to conveniently and rapidly evaluate multiple contaminants on site. In this study, with the assistance of ultrasound, we quickly determined the total PAEs in soft drinks using 1.5 mL of petroleum ether in one step. Then, we determined the characteristic molecular fluorescence spectrum of all PAEs in samples (excitation (Ex)/emission (Em) at 218/351 nm) using selectively concentrated sulfuric acid derivatization. The relative standard deviations of the fluorescent intensities of mixed solutions with five different PAEs were lower than 7.1% at three concentration levels. The limit of detection of the proposed method is 0.10 μmol L^−1^, which matches that of some of the chromatographic methods, but the proposed method uses less organic solvent and cheaper instruments. These microextraction devices and the fluorescence spectrometer are portable and provide an instant result, which shows promise for the evaluation of the total level of PAEs in beverages on site. The proposed method successfully detected the total level of PAEs in 38 kinds of soft drink samples from local supermarkets, indicating its potential for applications in the packaged beverage industry.

## 1. Introduction

The consumption of packaged food products is rapidly increasing. Public consumption of packaged water and soft drinks is widespread at work, at home, and while traveling. Consequently, the risk of exposure to many kinds of food contaminants through these beverages is higher than before, which is accordingly increasing healthcare costs. This poses a food analysis challenge, as an appropriate detection strategy is needed to identify potential contaminants. Advanced chemical analytical instruments such as chromatography, mass spectrometry, nuclear magnetic resonance, and spectroscopy can be used to accurately measure many hazardous compounds in food. However, these methods require complex and expensive instruments that are unsuitable for multi-target and large-scale analysis on site and in the field.

Phthalate esters (PAEs) are a group of endocrine disruptors that cause irreversible damage to the health of humans and other animals [1,2]. With increasing exposure to plastic fragments and microplastics, the environmental hazard posed by these ubiquitous PAEs is becoming increasingly worse [3]. The concentrations of 16 PAEs were found to vary from 35.9 to 36,225.2 ng g^−1^ in Yangtze River estuary sediment samples, and the dominant PAEs were di-isobutyl phthalate (DIBP), dibutyl phthalate (DBP), dioctyl phthalate (DEHP) [4]. PAEs were found in whole samples from 10 drinking water bottles in different levels, and the total PAE level was 81.4–358.7 ng·mL^−1^ [5]. Many kinds of PAEs exist in drinking water, thereby contributing to PAE pollution in beverages [6,7,8,9]. In addition, PAEs in sediment is closely associated to leakage from plastic fragments and microplastics [4]. Additionally, during storage and transportation, PAEs can be released, migrate, or evaporate into package contents, which is another important source of drink contamination [10]. At present, packaged commercial drinks are used daily worldwide. These commercial drinks are an important source of human exposure to PAEs, showing that the current chromatographic detection methods are unable to detect so many samples in time [11,12]. As such, large-scale, on-site, and rapid screening is required to deal with this situation. In addition, heavily contaminated samples identified in the screening step can be further analyzed by additional methods such as chromatography techniques to provide information so solutions can be created for this food analysis challenge. However, we could find no such method on the SCI database.

Extraction is a key step in any analytical method. PAEs are oil-soluble pollutants dispersed in water-soluble beverages. As such, dispersive liquid-liquid microextraction (DLLME) is a suitable strategy for extracting PAEs from nonalcoholic drinks, as it is regarded as a powerful pretreatment technique for trace amounts of fat-soluble target compounds in aqueous matrices. However, its extractants are highly toxic and dispersed in the environment [13]. According to our previous results, ultrasound and DLLME are suitable methods for this task, with some modifications, due to their several advantages, such as less toxicity, rapidity, low cost, strong enrichment capacity, and ease of operation [1,14]. In 2017, we found that our proposed method, ultrasound-assisted upper liquid-liquid microextraction (UAULLME), enables convenient extraction and collection, is almost nontoxic, and produces no residual solvents from the target compound solution [15].

Several techniques have been developed to detect PAEs [10,11,12,16]. From our previous work, we found that chromatography techniques are the preferred methods of determining PAEs due to their accurate and stable results [1]. However, if these methods are employed to determine PAE contamination, GC/HPLC with UV/MS exhibits some disadvantages: tedious procedures, requiring large amounts of hazardous organic solvents, expensive to perform in a special laboratory, and necessitating professional operators [17]. Additionally, these instruments cannot be easily transported and run on site. As an alternative, fluorescence analysis can provide nearly equally high sensitivity and selectivity with less expensive instruments, requiring only several milliliters of a sample and small amounts of organic solvents. In addition, instantaneous detection is the most attractive advantage of fluorescence analysis. Furthermore, given the general contamination of PAEs of a large number of food products, fluorescence analysis can be used on site to effectively overcome the weaknesses of chromatography techniques mentioned above.

As such, in this study, we constructed a facile and selective petroleum ether microextraction procedure combined with derivative fluorescence spectrometry to detect the total level of PAEs. We evaluated the total level of PAEs in soft drinks at both production and sale sites, finding that this method is rapid, low cost, easy to operate, low in toxicity, and provides miniaturized detection procedures. Our aim in this study was also to provide one method to deal with a major food analysis challenge.

## 2. Results

### 2.1. Optimization of the Petroleum Ether Microextraction Conditions

We tested two low-density and -toxicity solvents, i.e., petroleum ether and n-hexane. We recovered more of the more volatile solvent, petroleum ether, than n-hexane, at 58.6% and 32.8%, respectively, which was a significantly higher recovery (*p* < 0.05). Compared with n-hexane, the structures of PAEs are more similar to those of petroleum ether than to those of n-hexane. Therefore, we selected petroleum ether as extraction solvent in the following experiments.

In this petroleum ether microextraction method, the volume of the extraction solvent is an important factor, as it can influence both the formation of a cloudy mixture and extraction efficiency. Therefore, using DBP as a representation of the PAEs, we used different volume of petroleum ether, ranging from 0.50 to 2.00 mL, as the extraction solvent. As shown in Figure 1a, we observed a notably increase in PAE recovery from 0.50 to 1.50 mL. We obtained the highest recovery at 1.5 mL. At lower volumes, such as 0.50 mL, a stable cloudy solution did not form due to its high volatility. The recovery slightly decreased as the petroleum ether volume increased from 1.50 to 2.00 mL. The same result was observed by Jing et al. [18] and Zheng et al. [19]. As the volume of the extraction solvent increased, the analytes diluted in the extraction solvent phase, which decreased the sensitivity. Therefore, we selected 1.50 mL of petroleum ether as the optimal solvent for the next steps.

We then considered different treatment times, i.e., 0, 1, 5, 10, 15, 20, 25, and 30 min, to optimize the recovery. As shown in Figure 1b, the recoveries substantially increased as the ultrasonic time increased from 0 to 20 min, and the highest recovery was 89.5%. Then, we observed a decrease in recovery with increasing the ultrasonic time from 20 to 30 min. We previously observed a similar effect for SFO-DLLME with Sudan I and IL-DLLME for DBP with increasing ultrasound exposure time [1,14]. During petroleum ether microextraction, ultrasonic treatment produces and maintains the cloudy state in extraction mixtures, which is necessary for target analytes to come into contact with the extraction solvent. Consequently, extraction efficiency increased as the ultrasound exposure time increased. However, ultrasonic treatment might degrade the target analytes during extraction [20]. Therefore, we selected 20 min as the optimized treatment time for the following detection.

### 2.2. Optimizing Derivation Procedures

The fixed excitation and emission wavelength of a target compound is the qualitative basis used for fluorescence measurement. First, we determined the optimal volume of H_2_SO_4_ (98.3%) by considering 12 levels ranging from 0 to 8.00 mL. The maximum emission wavelength and fluorescence intensity of the derived standard solution are displayed in Figure 2a,b. Figure 2a shows that the maximum emission wavelength varied from 337 to 351 nm as H_2_SO_4_ (98.3%) volume increased from 0 to 4.00 mL. When the volume of H_2_SO_4_ (98.3%) was more than 4.00 mL, the maximum emission wavelength tended to stay at 351 nm. This suggests that 4.00–8.00 mL of H_2_SO_4_ (98.3%) can ensure a stable derivation product. We observed a sharp increase in the fluorescence intensity as the volume of H_2_SO_4_ increased from 0 to 4.00 mL, which then moderately increased until 8.00 mL, as shown in Figure 2b. The results show that the higher the concentration of sulfuric acid, the stronger the derivation and dehydration capacity. The fluorescence intensity did not reach a peak at 8.00 mL of H_2_SO_4_ (98.3%). In addition, the 2.00 mL water we used in the experiment instantaneously generated heat when mixed with H_2_SO_4_ (98.3%), which might have accelerated the reaction. Finally, we determined that 8.00 mL was the optimal volume of H_2_SO_4_ (98.3%) for this process, i.e., the ratio of water to H_2_SO_4_ (98.3%) was 1:4 (*v*/*v*) in the next step.

We also investigated the effects of the derivative reaction time (0, 10, 20, 30, and 40 min) and temperature (25 °C and 80 °C in water bath) on the fluorescence intensity of the derived target compounds. As shown in Figure 2c, the instrument response values notably increased as the reaction time increased from 0 to 30 min. Beyond 30 min, the instrument response values decreased because the derivation reaction needs sufficient time, but prolonged exposure to the air might change the product. We achieved similar results at 25 °C and 80 °C: we observed 11% higher fluorescence intensity at 80 °C than at 25 °C. To achieve maximum sensitivity, the temperature of the reaction system was kept at 80 °C for 30 min, and then dropped it to room temperature. Therefore, we chose 30 min and 80 °C in a water bath as the optimal conditions for further experiments.

### 2.3. Derivate Fluorescence Determination

#### 2.3.1. Optimization of the Fluorospectrophotometer Slit Width

In optimal test above, the intercept of the work curve was large, up to more than 400, which might have affected the linear range of the standard solutions. Therefore, we tested the effect of slit widths on these linear characteristics using a series of DBP solutions (seven levels within 0–10.00 µmol L^−1^). We determined the standard curve equation for the slit width pairs of 5/5, 1.5/10, 1.5/15, 1.5/20, 3/5, 3/10 nm/nm for Ex/Em, respectively, and drew the working curves of each pair. Those correlation coefficient and intercept are illustrated in Table 1. Table 1 shows that the minimum intercept and the best correlation coefficient were obtained at 3/5 nm/nm Ex/Em. Therefore, we selected a 3/5 nm of slit width as the optimal instrument parameter.

#### 2.3.2. Derivative Fluorescence Characteristics of Mixture of PAEs

The derivative fluorescence spectrum of each PAE member was not unknown. As such, in this study, we used five highly toxic PAEs that have attracted considerable attention in the national standards and the literature, i.e., DMP, DBP, DIBP, DEP, and DEHP, as representative of the total PAEs levels. We carefully investigated and compared their fluorescence characteristics, namely fluorescence emission wavelength, intensity, and linearity, using individual and multicomponent standard solutions containing equal concentrations at three total concentration levels, 0.25, 2.50, and 10.00 µmol L^−1^ The fluorescence emission wavelength and intensity of samples with individual and mixed PAEs are displayed in Figure 3. Figure 3 shows that highly consistent maximum emission wavelengths at 351 nm and 218 nm were excited for individual and mixed samples at different concentrations, respectively. In addition, the fluorescence intensity increased at the same rate as the PAE concentration for both individual and mixed solutions. The differences in the fluorescence intensity of the five individual compounds and the mixed PAEs were not significant (*p* > 0.05), and the RSD% were 3.74%, 7.03%, and 3.74% of for the mixed solutions at 2.50, 5.00, and 10.00 µmol L^−1^, respectively. This finding confirmed the highly consistent fluorescence performance for different individual PAEs and their mixtures, showing that the total level of PAEs can be obtained using the proposed method in one step.

### 2.4. Method Validation

#### 2.4.1. Linearity, Limit of Detection, and Limit of Quantification

We determined the linearity of the proposed method for the total level of PAEs by calibration using a series of mixed standards solutions containing equal concentrations for a total concentration of 0–20.00 µmol L^−1^. We observed the linearity of the mixed standards’ addition curves at 0.25–20.00 µmol L^−1^, which showed satisfactory linear correlation (R^2^ = 0.987).

According to IUPAC, we determined the limits of detection (LOD) and quantification (LOQ) from the analysis of seven replicate samples containing the target analytes at a level near the estimated detection limit at a 95% confidence level [15]. The LOD and LOQ of the proposed method are 0.10 μmol L^−1^ and 0.25 μmol L^−1^, respectively.

#### 2.4.2. Precision and Accuracy

We evaluated precision to assess the repeatability of the fluorescence spectrometer by measuring the intraday relative standard deviation (RSD) and the RSD of two consecutive days of nine repeated measurements. The RSD values were 2.2%, 4.4%, and 4.1% for 0.25, 2.50, and 5.00 μmol L^−1^ standard solutions containing single and mixed analytes in five equal concentrations, respectively.

To assess the accuracy of the method, we calculated the total recovery of analytes in real sample media using spiked five-component standard solutions of PAEs containing equal concentrations of each PAE in four kinds of soft drinks before the degassing step. The mean recoveries of total level of PAEs ranged from 81 to 103%. The RSD of triplicate measurements of the spiked samples at three concentration levels ranged from 3.3 to 10.1%. Our findings meet the requirements set by the current European legislation (the recovery values between 70% and 120%, RSD lower than 20%). Table 2 illustrates the results.

### 2.5. Evaluation of Total PAEs in Commercial Soft Drinks

We employed the proposed method to detect the total level of PAEs in 39 samples of soft drinks from six categories, aided by a series of five-component standard solutions with equal concentrations of components. As displayed in Table 3 only one carbonated beverage sample exhibited a negative result, and the total level of PAEs in all the other samples were less than 7.45 μmol L^−1^. The total PAE level in the mineral water samples was 2.91–2.30 μmol L^−1^, which was higher than that of purified water. Our results are similar to those reported by Huang and coworkers [5]. These results indicate that the water source used to produce these drinks might have been contaminated by PAEs, which is an important consideration in terms of the safety and health effects of natural mineral water drinks. However, the large difference in the total level of PAEs might have been due to their leaking from packages into the contents during the storage and transport of drink products, as well as other materials in soft drinks.

To understand the degree of and the differences in PAE migration into drink contents, we investigated the difference in the total PAE level between the same kinds of fruit juices in bottles in four pairs of juice samples. The total PAE contents were 1.5 to 5 times higher in juice in plastic bottles than in cans. Plastic bottles contain more plastic additives than the internal coatings of metal cans. Our results indicated that PAEs migrate into their contents, so plastic bottles are a potential media that may cause PAE contamination in soft drink products [1,17].

Table 3 confirms that we observed the highest level of total PAEs in fruit juices in plastic bottles. Detailed data are provided in Figure 4. In some juice samples, such as orange juices, peach and pomelo juices with honey, pear juices with sugar, and so on, we found high total PAE levels. Wu et al. [21] also monitored the level of four PAEs, i.e., DAP, DPRP, BBP, and DBP, in some bottled juice samples. They found only 1.85 μg L^−1^ DBP and 1.57 μg L^−1^ BBP in bottled pineapple juice samples, and no PAEs in bottled pear juice samples [21,22,23]. In addition, the second highest value was found in beverages with vitamin C. All the results indicated that pH or some complex natural ingredients in fruit juice might promote the migration of PAEs [21].

Subsequently, we studied the effect of storage on the PAE levels. We analyzed the total PAE levels in two pairs of samples with a manufacture date 7 months apart. However, the total level of PAEs was not significantly different between them. We also obtained this result in our previous study, which might have resulted from a lack of shaking or impact during storage [1]. In future studies, more soft drink samples in different packages should be screened for the total level of PAEs both at manufacturing and sale sites.

## 3. Materials and Methods

### 3.1. Chemicals, Materials, and Standards Preparation

We obtained dimethyl phthalate (DMP, ≥99%), dibutyl phthalate (DBP, 99%), di-isobutyl phthalate (DIBP, 99%), diethyl phthalate (DEP, 99.5%), and bis (2-ethylhexyl) phthalate (DEHP, ≥99%) from Sigma-Aldrich (Darmstadt, Germany). We purchased HPLC-grade petroleum ether and n-hexane from Merck Chemical Company (Darmstadt, Germany). We obtained analytical-grade H_2_SO_4_ (98.3%) from Shengqinghe Chemical Company (Nanjing, China). In addition, dealing with concentrated sulfuric acid and use special acid-resistant gloves and glasses

We used ultrapure water purified using a Millipore Purification System (Burlington, NJ, USA) in all experiments. To minimize the risk of contamination, we used glassware in all experiments, which we strictly washed and checked to ensure the lack of contamination with PAEs before use.

We prepared individual standard stock solutions of each phthalic acid ester (100 mmol L^−1^) and stored them at 4 °C. We prepared serial dilutions of individual standard stocks and multicomponent solutions containing equal concentrations (100 µmol L^−1^) of each compound on site by diluting the stock solution with methanol. We constructed calibration curves by plotting the means of triplicate measurements of peak highs against the concentrations of each compound.

### 3.2. Apparatus

We used an ultrasonic cleaner (KQ600E, Kunshan Ultrasonic Instrument, Jiangsu, China) for degassing the beverages and assisting with the emulsification in the upper microextraction procedure. We measured all excitation and emission spectra of samples with a fluorescence spectrophotometer (RF-5301, Shimadzu, Japan) equipped with 1 cm quartz cell.

### 3.3. Sample Collection and Pretreatment

We purchased 39 soft drinks from 6 categories from markets in Dongguan (China). We stored the unopened packages at room temperature. After we opened the packages, we immediately collected the samples in glass tubes, which we capped and stored at 4 °C until analysis. Before testing, we degassed the soft drink samples for 20 min using ultrasound irradiation.

### 3.4. Petroleum Ether Microextraction Procedures

We conducted petroleum ether microextraction following the procedure we previously described, with some modifications [1,14,15]. Briefly, we transferred 20.00 mL of a thoroughly degassed sample into a 25.00 mL glass tube with a narrow neck. We added less than 2 mL of low-density organic solvent. We capped the vial and subjected the contents to ultrasound for 20 min. Then, we added 6.0 g of anhydrous NaCl and vortex-mixed the vial for 1 min to obtain the saturated extraction mixture. We added several milliliters of PAE-free water to rise the liquid surface of a narrow-neck 3 cm flask. We kept the mixture still for 15 min to allow phase separation. Then, we collected the upper organic layer in the narrow neck. We extracted the lower layer of the extracted sample for another 2 replicates. We combined the extract in 25.00 mL glass tubes and removed the solvent using a nitrogen gas flow at room temperature. We redissolved the residue into 50 μL of methanol and added 1.00 mL of distilled water into the solution for fluorescence analysis, which we collected into glass tubes. We then capped the tubes and stored them at 4 °C until analysis. For each kind of beverage, we used samples after 3 extractions as controls.

During the optimization of the extraction conditions, we calculated recovery using the following equation:Recovery = 100% × (C_t_ − C_0_)/C_a_(1)
where C_t_, C_0_, and C_a_ are the concentration of PAEs in sample solution after adding a certain level of individual or five-component standard analytes, the PAE concentration in the original sample solution, and the added concentration of PAEs, respectively.

### 3.5. Derivation and Detection of PAEs

We quickly added 4 mL of H_2_SO_4_ (98.3%) that we obtained from the previous microextraction step to the extract. We vortexed the mixture for 1 min and then allowed it to cool to room temperature before fluorescence measurement was carried out. When the concentrated sulfuric acid made contact with a small amount of water, heat was generated, instantly producing identical molecular fluorescence spectra for all PAEs in the samples with excitation (Ex)/emission (Em) at 218/351 nm. Therefore, all kinds of PAEs in the sample could be identified using the proposed method.

In this step, based on the fluorescence emission wavelength and intensity values, we optimized several parameters, including the volume of H_2_SO_4_ (98.3%) (0–8 mL), temperature (25 °C and 80 °C in water bath), time (0, 10, 20, 30, and 40 min) of the derivation reaction, with 5/5 nm slit widths using a DBP single standard at a concentration of 0.4 µmol L^−1^.

### 3.6. Fluorescence Spectrum Analysis

#### 3.6.1. Optimization of Fluorospectrophotometer Parameters

We optimized the slit widths (5/5, 1.5/10, 1.5/15, 1.5/20, 3/5, 3/10, nm/nm for Ex/Em, respectively) of the fluorospectrophotometer to obtain the best possible resolution, sensitivity, and appropriate response value using a series of series DBP solutions with concentrations of 0–1.00 µmol L^−1^ in 7 levels).

#### 3.6.2. Consistency of the Fluorescence Characteristics of Diverse PAEs

We investigated the consistency of the fluorescence emission wavelength and intensity of different PAEs using 0.25, 2.50, 5.00, and 10.00 µmol L^−1^ standard solutions of individual compounds and multicomponent solutions containing equal concentrations of the compounds.

#### 3.6.3. Fluorescence Intensity Detection

According to the results of the above optimization, we employed the optimal conditions for fluorescence intensity detection by RF-5301 PC. We fixed the scan rate at 600 nm min^−1^. The optimal excitation and emission wavelengths of the PAEs were 281 and 350 nm, respectively. All measurements were recorded at room temperature in triplicate.

### 3.7. Statistical Analysis

We collected and analyzed all data using Microsoft Excel (Microsoft Office, Redmond, WA, USA). Each test contained three parallel samples. We considered differences among the mean values as significant if their statistical *p*-value was <0.05.

## 4. Conclusions

In this study, we developed a rapid, facile, selective, low-cost, low-toxicity, easy-to-operate method for determining the total level of PAEs in soft drinks: molecular fluorescence based on petroleum ether microextraction and derivation by H_2_SO_4_ (98.3%). After optimizing the conditions of the assay procedures, i.e., microextraction, derivation, and slit width of the instrument, we found that five phthalates (DMP, DBP, DEP, DIBP, and DEHP) in individual and mixed solutions in equal concentrations displayed consistent fluorescence spectral characteristics at 218/351 nm for Ex/Em, respectively. We validated the proposed method, which achieved satisfactory precision and accuracy. We found that the relatively low LOD of the proposed method is similar to that of chromatographic techniques. We successfully applied the proposed method to the detection of the total level of PAEs in 39 samples from six categories of soft drinks. The proposed method has the potential to be used for the on-site monitoring of the contamination of soft drinks in terms of the total level of PAEs.

## Figures and Tables

**Figure 1 molecules-27-04157-f001:**
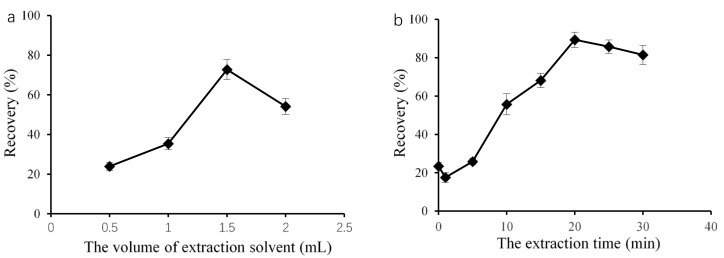
Effect of the petroleum ether microextraction conditions on target recovery: extraction solvent volume (**a**) and the extraction time (**b**).

**Figure 2 molecules-27-04157-f002:**
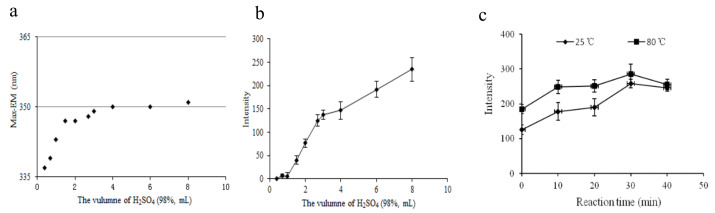
Effect of conditions on the fluorescence characteristics: (**a**) volume of H_2_SO_4_ (98%) on the maximum emission wavelength, (**b**) volume of H_2_SO_4_ (98%) on the fluorescence intensity, (**c**) reaction time and temperature on the fluorescence intensity of the aqueous DBP solution.

**Figure 3 molecules-27-04157-f003:**
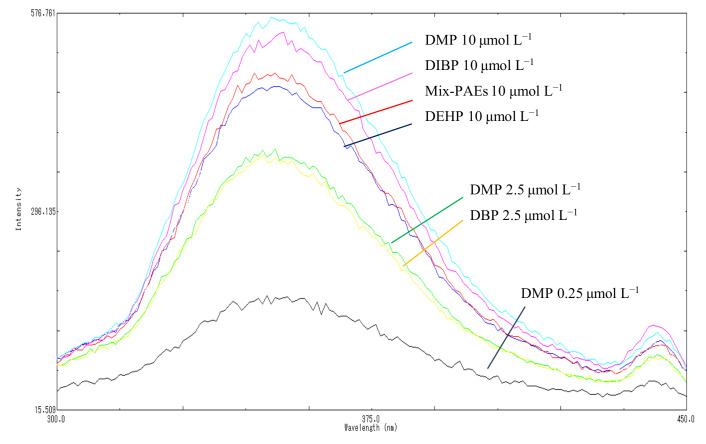
Fluorescence emission spectrum of individual and mixed solutions.

**Figure 4 molecules-27-04157-f004:**
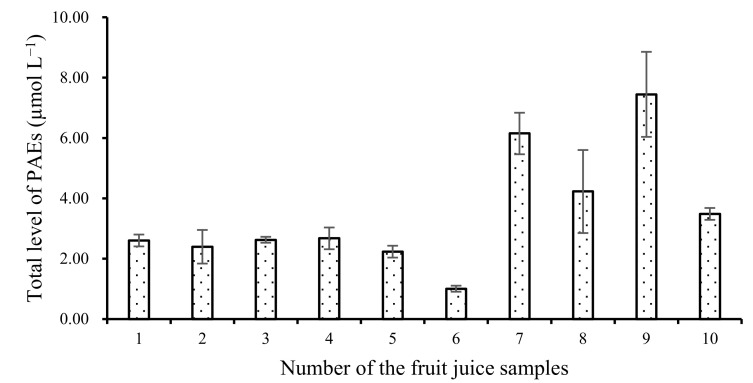
Total level of PAEs in each fruit juice sample in plastic bottle.

**Table 1 molecules-27-04157-t001:** Effect of slit width on the linear characteristics of the series DBP solution.

Slit Width (Ex/Em, nm)	R^2^	Intercept
5/5	0.9757	408
1.5/10	0.9366	177
1.5/15	0.9519	346
1.5/20	0.9570	570
3/5	0.9997	126
3/10	0.9792	437

**Table 2 molecules-27-04157-t002:** Total concentrations and recovery of PAEs in spiked soft drink samples using five-component standard solutions containing equal concentrations of the components.

Sample	Added (µmol L^−1^)	Mean Relative Recovery (%)	RSD (%)
Iced black tea in bottle	1.60	86	6.8
Vitamin water in bottle	1.60	81	4.4
Cold brewing tea in bottle	3.20	103	10.1
Mineral water in bottle	1.00	98	3.5

**Table 3 molecules-27-04157-t003:** The total PAE levels in common beverages (µmol L^−1^).

Sample	Contain (Mean)	Median Value
Carbonated beverage (*n* = 5)	ND–3.18	2.86
Tea drink (*n* = 5)	1.91–4.78	2.72
Fruit juice in bottle (*n* = 10)	1.01–7.45	2.65
Fruit juice in can (*n* = 4)	1.52–3.39	2.61
Purified water (*n* = 2)	1.84–3.25	-
Mineral water (*n* = 8)	2.91–3.30	3.20
Function drink (*n* = 5)	1.21–5.75	2.23

Note: ND, not detected.

## Data Availability

Not available.

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
