# Peer review of "Facile and Selective Determination of Total Phthalic Acid Esters Level in Soft Drinks by Molecular Fluorescence Based on Petroleum Ether Microextraction and Selective Derivation by H2SO4"

_molecules, 2022, doi:10.3390/molecules27134157_

Round 1

Reviewer 1 Report

In this manuscript, the authors report the use of 98.3 % sulfuric acid assisted with heating at 80 °C for fluorescence derivatization of phthalic acid esters. The developed method has a lot of disadvantages and hazardous conditions.

1- The procedure for derivatization of PAE is very dangerous as it involves the use of 98% sulfuric acid and mixing it with water, then heating at 80 °C. This method is not safe for the analyst or to the laboratory. The handling of highly concentrated sulfuric acid needs very high caution. Besides, measuring highly concentrated sulfuric acid solution in a fluorometer is a very dangerous process, and any leakage of sulfuric acid may destroy the instruments.

2- Furthermore, the mechanism of the formation of the fluorescent derivative is not discussed at all. Moreover, the selectivity of the method for PAEs was not studied at all.

3- Additionally, the English of the manuscript needs to be improved.

At last, I think that a manuscript that involves such dangerous and hazardous reaction conditions is not suitable for publication.

Author Response

Responses to Reviewer 1

Q1- The procedure for derivatization of PAE is very dangerous as it involves the use of 98% sulfuric acid and mixing it with water, then heating at 80 °C. This method is not safe for the analyst or to the laboratory. The handling of highly concentrated sulfuric acid needs very high caution. Besides, measuring highly concentrated sulfuric acid solution in a fluorometer is a very dangerous process, and any leakage of sulfuric acid may destroy the instruments.

R1- Firstly, thank you for your earnestly reminding. In this manuscript, as 4 ml of concentrated sulfuric acid comes into contact with 1 ml of water, the mixed solution generates heat. Then, put the hot solution into the water bath and keep it at 80 degrees. This procedure is safe for the staff of the chemical laboratory. In addition, the sample tank is made of quartz, and it will not be damaged by the sulfuric acid, and leakage. Standardized operation and protective gloves can eliminate the occurrence of these accidents you mentioned. Compared with the production of a large amount of mixed organic waste liquid and long analysis time in other methods, we think this method is very environmental-friend and innovation. We also think this comment is very good because laboratory safety really needs high attention.

Q2- Furthermore, the mechanism of the formation of the fluorescent derivative is not discussed at all. Moreover, the selectivity of the method for PAEs was not studied at all.

R2-The sentence ‘When the concentrated sulfuric acid contacts with a small amount of water, it will generate heat, then a novel identical characteristic of molecular fluorescence spectrum of all PAEs in samples with excitation (Ex)/emission (Em) at 218/351 nm, respectively, was instantly achieved. From this experimental result, it was inferred that phthalic acid and phthalic anhydride were produced successively under the derivatization of concentrated sulfuric acid. The united fluorescence spectrum for all kinds of PAEs was excited by the rigid plane structure of phthalic anhydride. Therefore, all kinds of PEAs in the sample can be selected in this proposed method.’ Has been added in 3.5 part in Line 349-356.

Q3- Additionally, the English of the manuscript needs to be improved.

R3-We will send the manuscript to accept language editing service of MDPI.

Reviewer 2 Report

Dear Sir

Good evening, I am greatly appreciated for choosing me as a reviewer for the article entitled “Facile and selective determination of the total level of phthalic acid esters in soft drinks by molecular fluorescence based on  petroleum ether microextraction and selective derivated by  H2SO4” in your distinguished journal.

Decision: major revision

  1. There are several spelling and grammatical mistakes in the entire text. English needs to be polished by a proofreader.
  2. In abstract: You must clarify the abbreviation (PAEs, Em, Ex,LOD) in the first use.
  3. All figures are very poor in resolution and not clear. So, it must be replaced by others with high resolution to be suitable for publication.
  4. Why do you study only extraction at 25â—¦ and 80â—¦ in the study of the effect of temperature?
  5. All references in the text were written as numbers, but there are many references written in the alphabet as in lines 62, 82, 85, 219, and 279. So, they must be replaced by numbers.
  6. Table 4 must be reviewed again.
  7. The formula of sulphuric acid in line 143 must be corrected.

Author Response

Thank you for your comments very much.

Q1: There are several spelling and grammatical mistakes in the entire text. English needs to be polished by a proofreader.

R1: We will send the manuscript to accept the language editing services of MDPI.

Q2: In abstract: You must clarify the abbreviation (PAEs, Em, Ex, LOD) in the first use.

R2: The words ‘phthalic acid esters, Excitation (Ex), Emission (Em), limit of detection’ has been added before each abbreviation in abstract.

Q3: All figures are very poor in resolution and not clear. So, it must be replaced by others with high resolution to be suitable for publication.

R3: Figure 1,2,4 have been revised. But Figure 3 is the original image copied from the instrument. In order to maintain its authenticity, such images are generally provided in the literature, just like in the liquid chromatogram methods.

Q4: Why do you study only extraction at 25℃and 80℃ in the study of the effect of temperature?

R4: Your question is reasonable. But, the present results can also display information on the difference between room temperature and higher temperature.

Q5: All references in the text were written as numbers, but there are many references written in the alphabet as in lines 62, 82, 85, 219, and 279. So, they must be replaced by numbers.

R5: All the author name and publish time have been cancelled or replaced by numbers.

Q6: The formula of sulphuric acid in line 143 must be corrected.

R6: In the molecular formula of sulfuric acid, the subscript of the number has been modified.

Reviewer 3 Report

The manuscript described a method for determination of total content of phthalic acid esters in soft drinks, based on a simple extraction followed by derivatization and determination by fluorescence. The method is simple and economically affordable. However, it is not able to distinguish between each different phthalic acid ester. Additionally, the validation of the method was poorly design and executed and the method was applied in matrices not validated. Therefore, I do not recommend the manuscript for publication in Molecules. I have included some more specific comments below, in case authors would like to consider improving their study and manuscript.

English should be significantly and carefully improved through the whole manuscript. Some sections and paragraphs are really difficult to follow.

The advantages and usefulness of the method should be clearly stated.

Please specify the meaning of acronyms the firs time they appear (i.e. PAEs in the abstract).

Line 5 of the abstract: eliminate “in”

Please, specify in the abstract what do you exactly mean by “a novel identical characteristic molecular fluorescence spectrum”.

Line 71-80, the extraction presented in this study is not DLLME or UAULLME, so this paragraph and all those references needs to be deleted.

Iine 91-93: which kind of pollutant were monitored in those drinks? If those publications are not dealing with PAEs, please delete the paragraph and the references.

Line 108: why reference 1 is there? – please modify the text accordingly.

Line 112-115: this statement does not bring any added value, please delete it.

When optimizing the extraction conditions, what was use as recovery? – recovery for which analyte? Please specify clearly in the text.

In my opinion individual recoveries for each of the analytes included in the study need to be consider independently to avoid hiding different effects.

What is the reason why the recovery decreases as the extractant volume increased from 1.5 to 2 mL?

Line 179: please specify what is DBP.

Line 187: the aim of the study should be mentioned much earlier, i.e. in the abstract and the last section of the introduction.

Line 195-197: it is difficult to understand which error was reduced.

Section 2.4.2.:

Precision of the whole method, including real sample and sample preparation should be evaluated.

Please specify if the concentration level used for precision evaluation was for total PAEs or individual one.

Precision was estimated at 2.50 umol/L which is 10 times higher than the estimated LOQ, and accuracy was evaluated at concentration ranging from 1.0 to 3.2 umol/L, which is 4-13 times higher than estimated LOQ. Therefore, performance of the method at the LOQ level was not proved. Performance of the method should be validated at the LOQ concentration level.

Section 2.4.3.:

Comparison of the method should be done based on LOQ validated in terms of precision and recovery and not based on theoretical estimations of the LOD.

One important aspect that authors forgot to include in their comparison is the fact that chromatographic methods are able to quantify individual PAEs, which may be of extreme importance for some applications.

Section 2.5.:

The method was applied in matrices for which the method was not validated (i.e. carbonated drinks, fruit juices, function drink). In order to be able to trust obtained results the method should be properly validated on different matrices before its application.

Author Response

Thank you for your attention and your comments very much.

Q1: English should be significantly and carefully improved through the whole manuscript. Some sections and paragraphs are really difficult to follow.

R1: We will send the manuscript to accept the language editing services of MDPI.

Q2: The advantages and usefulness of the method should be clearly stated.

R2: The content has been thoroughly revised from line 59 to line 78, the advantages and usefulness of the method should be clearly stated.

Q3: Please specify the meaning of acronyms the first time they appear (i.e. PAEs in the abstract).

R3: The words ‘phthalic acid esters, Excitation (Ex), Emission (Em), limit of detection’ has been added before each abbreviation in abstract.

Q4: Line 5 of the abstract: eliminate “in”.

R4: The ‘in’ after ‘involved’ in line 5 of the abstract has been cancelled.

Q5: Please, specify in the abstract what do you exactly mean by “a novel identical characteristic molecular fluorescence spectrum”.

R5: We found that the same fluorescence spectrum produced by all kinds of PAEs whether individuals or mixtures. It will be identified later as we have not identify specific compound.

Q6: Line 71-80, the extraction presented in this study is not DLLME or UAULLME, so this paragraph and all those references needs to be deleted.

R6: The paragraph and references 15 and 16 have been canceled.

Q7: Iine 91-93: which kind of pollutant were monitored in those drinks? If those publications are not dealing with PAEs, please delete the paragraph and the references.

 R7: The sentence in Line 99-100 and the references 20-22 have been canceled.

 Q8: Line 108: why reference 1 is there? – please modify the text accordingly.

R8: The sentence ‘The appropriate extraction solvent and its volume of petroleum ether microextraction were optimized for achievement of high recoveries [1].’ has been canceled.

Q9: Line 112-115: this statement does not bring any added value, please delete it.

R9: In Line 112-115, the sentence Theoretically, the solvent must efficiently dissolve the analytes should not be so volatile to be evaporated during the extraction step.’ has been deleted

Q10: When optimizing the extraction conditions, what was use as recovery? – recovery for which analyte? Please specify clearly in the text.

R10: The paragraph has been added, i.e., ‘During optimization the extraction conditions, recovery was calculated using the following equation:

Recovery =100% × (Ct – C0)/Ca

where Ct, C0, and Ca are the concentration of PAEs in sample solution after adding a certain level of individual and five-component standard analyzes, the PAEs concentration in original sample solution, and the added concentration of PAEs, respectively.’

Q11: In my opinion individual recoveries for each of the analytes included in the study need to be consider independently to avoid hiding different effects.

R11: We aimed to evaluate the total level of PAEs. So the recovery test was designed to display the efficience in total mixture determination.

Q12: What is the reason why the recovery decreases as the extractant volume increased from 1.5 to 2 mL?

R12: The sentences in line 128-131 have been revised as ‘On the other hand, the recovery displayed a slight decrease as the volume increased from 1.50 mL to 2.00 mL The same result was observed by Zheng et al. [18] and Jing et al.,[19]. The reason was inferred that as increasing the volume of the extraction solvent, the analytes were diluted in the extraction solvent phase, which decreased the sensitivity.’

Q13: Line 179: please specify what is DBP.

R13: The ‘DBP’ has been defined in line 55 ‘Dibutyl Phthalate’.

Q14: Line 187: the aim of the study should be mentioned much earlier, i.e. in the abstract and the last section of the introduction.

R14: The sentence ‘The aim of the proposed method was obtained the total level of PAEs using the fluorescence detection’ in line 187 has been canceled.

Q15: Line 195-197: it is difficult to understand which error was reduced.

R15: The sentence ‘The error is reduced by three total levels, i.e., 0.25, 2.50, 5.00, 10.00 µmol L-1 for individual and mixed targets.’ has been revised as ‘To reduce the error, three total concentration levels, namely 0.25, 2.50, 5.00 and 10.00 µ mol L-1, were used for determination.’

Section 2.4.2.

Q16: Precision of the whole method, including real sample and sample preparation should be evaluated.

R16: Considering high chemical stability of PAEs and the samples have been degassed and highly salted, we did not perform precision test for the real sample. But the RSD% of Method Recovery can provide the information of the RSD% of analytes in sample, which is lower than 10.1%.

Q17: Please specify if the concentration level used for precision evaluation was for total PAEs or individual one.

R17: The phrase ‘standard solution contained single and mixture with five equal concentration component,.’ has been added in line 226-227

Q18: Precision was estimated at 2.50 umol/L which is 10 times higher than the estimated LOQ, and accuracy was evaluated at concentration ranging from 1.0 to 3.2 umol/L, which is 4-13 times higher than estimated LOQ. Therefore, performance of the method at the LOQ level was not proved. Performance of the method should be validated.

R18: The RSD% at 0.25 umol/L of mixed targets was 2.2%, which has been added in line 225-226.

Section 2.4.3.:

Q19: Comparison of the method should be done based on LOQ validated in terms of precision and recovery and not based on theoretical estimations of the LOD.

R19: Table 3 and its content have been canceled.

Q20: One important aspect that authors forgot to include in their comparison is the fact that chromatographic methods are able to quantify individual PAEs, which may be of extreme importance for some applications.

R20: This content has been highlighted ,in line 59-80 of introduce again.

 Section 2.5.:

Q21: The method was applied in matrices for which the method was not validated (i.e. carbonated drinks, fruit juices, function drink). In order to be able to trust obtained results the method should be properly validated on different matrices before its application.

R21: The sentence ‘For each kind of the beverage, samples after 3 times extraction was used as the control for itself.’ has been added in line 338-339.

Reviewer 4 Report

See attached file

Author Response

Thank you for your comments very much.

Q1: Extensive English revision. The manuscript is written using an informal English style. R1: We will send the manuscript to accept the language editing services of MDPI.Q2: In the abstract the authors describe the petroleum ether as a “low toxic solvent” however checking the Safety Datasheet, Petroleum ether is described as: “H315 Causes skin irritation, H320 Causes eye irritation, H336 May cause drowsiness or dizziness, H401 Toxic to aquatic life, H411 Toxic to aquatic life with long-lasting effects”. For this reason, I suggest modifying the sentence and adding a comparison between other solvents used in different methods. R2: The phraselow toxic solventhas been cancelled. Sorry, this sentence was based on the fact that petroleum ether is low toxic compared with other organic solvents such as methanol and acetonitrile. And it is really not rigorous enough.Q3. The authors describe the optimization of the method, in the results from section 2.1 to section 2.3.2. However, this description is only qualitative, and no statistical support is given. I suggest adding more quantitative data and reporting the comparison by applying the statistical test. For instance: the amount of the extracts obtained with petroleum ether and n-hexane (at least 3indipendet extraction for each solvent) should be compared using a t-test. The same statistical test should be applied to the selection of volume, time of extraction, and all the remained parameters.

R3: The steps of applying statistical test to full parameter optimization are more stringent in statistics, indeed. However, statistical tests can be omitted in analytical chemistry, which can be found in more advanced journals, such as ‘Food Chemistry 377 (2022) 131980, DOI 10.1016/j.foodchem.2021. 131980’ and ‘Sensors and Actuators b-Chemical 356(2022)131325; DOI: 10.1016/j.snb.2021.131325’, etc. Therefore, only the phrase ‘which exhibited higher recovery with significant diference (P<0.05)’ has been added in line 120.

Q4. The comparison with the methods already reported in the literature it is slightly superficial. To compare two analytical method the same samples should be analysed using the two methods and statistical analysis of the results should be performed. In detail: the authors should perform the analysis of the 39 kinds of soft drinks applying their method and one (or all, depending on time and costs) of the reported methods in Table 3. After the statistical comparison of the results, the authors could write if their method is “better” or “ give the same results, but is less time consuming” etc. than the already published methods.R4: We think your comments is reasonable and professional, though the same strategy like this is still performed in many more advanced magazines, such as Table 1 from Wu et al (Food Chemistry, 2020, 333, 127532, https://doi-org.tue.80599.net/10.1016/j.foodchem.2020.127532) . We did not design that experiment step yet. Therefore, the Table 3 has been totally canceled.

Reviewer 5 Report

Manuscript Number: 1665167

Title: Facile and selective determination of the total level of phthalic acid esters in soft drinks by molecular fluorescence based on petroleum ether microextraction and selective derivated by H2SO4

The manuscript is correctly written, scientifically sound, and clearly presented in good written English. Employed experimental methods are adequate, clear, and complete to allow repetition of the work. Data are properly interpreted to support the conclusions. Relevant issues in the discussion are adequately discussed.

This is an interesting manuscript introducing a novel analytical method for the detection of phthalic acid esters in common beverages using molecular fluorescence and microextraction.

Major concerns:

Limitations of their approach: The authors should include a discussion of the limitations of the method, particularly compared to existing methods. Is there limited sensitivity? Limited pH range? Potential for interfering contaminants? The authors also need to compare the performance of their method with other reported methods in terms of sensitivity, the limit of detection, etc. I suggest inserting a comprehensive table for that purpose.

The work proposed is satisfactory and should be considered for publication. As a way of improving it, I'd suggest:

  • adding a comparison of the results obtained to those described in the literature and standard method.
  • a better description of the spiking of real samples. At which point was the sample spiked?

Minor concerns:

Line 47 & 175

Font

Line 143, 180, 196, 204, 363 – super- and subscripts

H2SO4, μmol L-1 at 7 levels, min-1

Line 225 & 235

five-conponent?

Check the whole text for omitted space.

Author Response

Thank you for your attention and your comments very much.

  • Q1: adding a comparison of the results obtained to those described in the literature and standard method.

R1: Now, the proposed method in the Table 3 has been totally canceled now. According reviewers’ comments, we redefined the significance of the proposed method in introduce. It is a quickly and preliminary screening strategy before chromatographic method for the heavy pollution samples. The purpose of this method is strengthening on-site supervision without affecting commodity circulation. Therefore, two methods are complementary.

Q2: a better description of the spiking of real samples. At which point was the sample spiked?

 R2: The phrase ‘before degasing step’ has been added in line 230 of second paragraph.

Minor concerns:

Q3: Line 47 & 175

R3: The font for full text has been revised as ‘Palatino Linotype and 10’, including Line 47 & 175.

Q4: Line 143, 180, 196, 204, 363 – super- and subscripts (H2SO4, μmol L-1 at 7 levels, min-1)

R4: All the super- and subscripts of H2SO4 and units has been corrected.

Q5: Line 225 & 235 five-conponent?

R5: All the phrase ‘five-conponent’ has been corrected as ‘five-component’.

Q6: Check the whole text for omitted space.

R6: All the omitted spaces have been checked and added.

Round 2

Reviewer 1 Report

The authors have revised their manuscript; however, their revision and answers are not satisfactory. The manuscript should be re-revised in light of the following comments:

1- A section called precautions should be added to the manuscript. In this section, the authors should mention that the analyst should pay high caution during dealing with concentrated sulfuric acid and use special acid-resistant gloves and glasses.

2- Also, the method is not environmentally green at all. The authors should remove any green description for their method from the manuscript.

3- From Fig. 2C, using an 80 °C with concentrated sulfuric acid is very dangerous, and at the same time, it does not provide a significant increase in fluorescence intensity (after 30 min). Hence, the authors should state that they have used 80 °C to achieve maximum sensitivity; however, it is advisable to use room temperature for more safety.  

4- The authors claimed that the formed fluorescent products are phthalic acid and phthalic anhydride. They did not supply any supporting data for their claim. They should study the formed fluorescent product with mass spectrometry and NMR analysis. Also, the fluorescence spectra of phthalic acid and phthalic anhydride should be provided, and the similarity of the spectra features with those of the spectra in Figure 3 should be discussed.

Author Response

Thank you for your help.

Reviewer 2 Report

For my part, I have no further comments to make and the paper is suitable for publication.

Author Response

Thank you for your attention.

Reviewer 3 Report

My major comment still remains: the method is not fully validated yet in the matrices for which it was applied for. Some other comments are still pending to be clarified. Please, see below:

English should still be improved further.

Please, specify in the abstract what do you exactly mean by “a novel characteristic molecular fluorescence spectrum”. This is not understandable. The fluorescence spectrum of a substance can´t be novel.

Line 65-66: USA and Chinese regulation limits consider individual PAEs or total quantity? Please, specify I the text. If they consider individual compounds, please specify how this method helps to comply with the legislation.

Section 2.1.

Line 107 – 110: not an understandable paragraph. Please, rephrase.

As requested in previous review, When optimizing the extraction conditions, recovery for which analyte was consider to evaluate the efficiency of the extraction? Please specify clearly in the text.

The reason given for recovery decrease as the extraction solvent increase from 1.5 – 2 mL (dilution and decrease in sensitivity) is not convincing. First, this effect is not produced when increasing the volume from 0.5 to 1.5 mL, which involves also dilution. Secondly decrease in sensitivity does not imply decrease in recovery (signal in the extract should be compared with a standard diluted at the same level).

Section 2.3.2.

Which error is reduced by using 3 concentration levels? – I do not understand which kind of error can be reduced by using 3 total concentration levels.

2.4.1.

LOQ level should be experimentally confirmed by validating the method (trueness and precision) at this concentration level in real sample. Please note that the trueness of the method was evaluated at concentration ranging from 4 to 12.8 times higher than the LOQ concentration level.

Author Response

Thank you for your help.

Reviewer 4 Report

The authors follow the suggested revision's or decided to remove the sections that arose the major concernes. I suggest accepting the manuscript  after this two minor furhter revision:

  • At line 260 the authors write "All the  results indicated that pH or some complex natural ingredients in fruit juice might promote the migration of PAEs". This sentence needs a literature reference or a further discussion to be acceptable.
  • Considering that Table 3 was deleted, the authors should rename Table 4 as Table 3 along the manuscript.

Author Response

Thank you for your help.
